# Myxoid Stromal Histophenotype Is Associated with High-Grade and Persistent Cervical Intraepithelial Neoplasia

**DOI:** 10.3390/pathophysiology32040055

**Published:** 2025-10-13

**Authors:** Leila Stabayeva, Madina Mergazina, Yevgeniy Kamyshanskiy, Gulchekhra Ikhtiyarova, Zhanna Amirbekova, Gulnazira Imanbayeva, Olga Kostyleva

**Affiliations:** 1Department of Morphology, Karaganda Medical University, Karaganda 100008, Kazakhstan; mergazina@qmu.kz (M.M.); amirbekova@qmu.kz (Z.A.); imanbaeva@qmu.kz (G.I.); kostyleva@qmu.kz (O.K.); 2Department of Pathology, Clinic of Karaganda Medical University, Karaganda 100000, Kazakhstan; kamyshanskiy@qmu.kz; 3Department of Obstetrics and Gynecology, Bukhara State Medical Institute, Bukhara 200118, Uzbekistan; ixtiyarova7272@mail.ru

**Keywords:** cervical intraepithelial neoplasia, extracellular matrix of the cervix, stromal remodeling, collagen, HPV, histophenotype

## Abstract

**Objectives:** To evaluate the diagnostic and prognostic value of histophenotyping of the extracellular matrix of the cervical stroma at cervical intraepithelial neoplasia (CIN). **Methods:** Retrospective analysis of 160 biopsies and surgical preparations of the cervix in women of reproductive age included cases of CIN 1–3 and the group with confirmed persistence or lesion progression (CIN P) at repeated biopsy. The control group (*n* = 40) consisted of morphologically intact cervical tissue. Histophenotypes were evaluated by staining with hematoxylin, eosin, and Masson trichrome, and classified as follows: normal (dense parallel bundles of type I collagen), intermediate (disorganized and fragmented type I collagen fibers), and myxoid (amorphous weakly fibrillar matrix). The clinical, viral, and inflammatory characteristics between histophenotypes were statistically compared. **Results:** The distribution of histophenotypes of the extracellular matrix of the cervix varied significantly depending on the CIN degree (*p* < 0.001). In the control group, the normal pattern was detected in 97.5% of cases; its frequency decreased from CIN 1 (27.5%) to CIN 2 (12.5%) and was absent at CIN 3. The frequency of the myxoid pattern increased significantly in severe and persistent forms: 55% at CIN 3 and 62.5% at CIN P. Human papillomavirus 16/18 was most frequently detected in groups with intermediate (69.1%) and myxoid (27.2%) patterns. Inflammatory changes were more often accompanied by disorganized extracellular matrix; however, intermediate and myxoid types also occurred in the absence of inflammation. **Conclusions:** The myxoid histophenotype of the extracellular matrix is significantly associated with the high degree of dysplasia and CIN persistence. It can reflect the morphological equivalent of tumor-associated stroma remodeling. Histophenotyping of the extracellular matrix of the cervix appears to be a promising method of risk stratification and may complement existing diagnostic algorithms for CIN.

## 1. Introduction

Cervical intraepithelial neoplasia (CIN) remains one of the most common forms of precancerous lesions of the cervix. CIN is a clinically significant but often diagnostically complex category of gynecological pathology. CIN has a high grade of morphological, colposcopic, and molecular heterogeneity, which makes it difficult to stratify risk early and justify individualized patient management tactics. In most cases, CIN 1 is prone to spontaneous regression within 12–24 months [1,2,3], whereas dysplasia of high grades (CIN 2 and CIN 3) is associated with the significantly higher risk of transformation into invasive cervical cancer [4,5]. In recent years, there has been a paradigm shift towards the organ-preserving personalized approach to the treatment of this pathology, especially in young women with preserved reproductive potential and the absence of high-risk factors [6].

Cases of persistent and recurrent CIN are of particular clinical importance. In these cases, neoplastic changes persist or reappear after treatment. Such forms can occur both in the presence and absence of persistent human papillomavirus infection (HPV) [7] and are often associated with an increased risk of therapeutic failure, re-progression, and malignant transformation [8,9,10]. Against this background, there is an increasing need for verified morphological and molecular markers that can improve the accuracy of predicting the clinical behavior of neoplasia and help in choosing a personalized follow-up strategy.

The majority of diagnostic approaches are focused primarily on epithelial changes, despite the development of cytological, colposcopic, and biopsy methods. At the same time, the subepithelial stroma remains significantly underestimated diagnostically. Meanwhile, the extracellular matrix (ECM) of the stroma is not only the architectural framework, but also the actively regulating element of tissue homeostasis. It controls proliferation, migration, apoptosis, and differentiation of epithelial cells by mediating signaling interactions between stroma and epithelium. Modifications of the extracellular matrix composition and mechanical properties during carcinogenesis are crucial for the onset and progression of a tumor [11,12]. In addition, the restructuring of the extracellular matrix is an important element of chronic inflammation and epithelial–stromal interaction, i.e., processes of critical importance for CIN persistence [13,14]. Accumulated data demonstrate that changes in the microenvironment of the cervical stroma include increased activity of matrix metalloproteinases, which leads to excessive degradation of collagen [15,16,17,18,19,20,21]. Such structural and molecular changes contribute to the preservation and prolongation of adverse changes, which can lead to stable changes in the extracellular matrix.

This study’s aim was the morphological assessment and comparative analysis of histological patterns of the extracellular matrix of the cervical stroma in normal conditions, at CIN of varying severity, as well as at the persistence or progression of the disease in repeated biopsies.

## 2. Materials and Methods

### 2.1. Tissue Samples and Selection of the Study Group

This multicenter retrospective cohort study consistently included biopsy and surgical cervical samples from women of reproductive age with cervical intraepithelial neoplasia (CIN) 1–3. The samples were obtained as the result of diagnostic or therapeutic intervention in Karaganda Regional Clinical Hospital, the Clinic of Karaganda Medical University and Multidisciplinary Center for Mother and Child of Temirtau (Kazakhstan), from January 2017 to December 2024. The control group (defined as the relative physiological reference) included cervical tissue samples obtained during routine gynecological procedures and autopsies without morphological signs of inflammatory, dysplastic, or neoplastic changes.

Study groups (CIN 1–3) and CIN P were formed according to the histology of the epithelium (dysplasia of stratified squamous epithelium) and the clinical course of CIN (progression/persistence). The CIN 1–3 groups included women of reproductive age (18–49 years old) who underwent the planned diagnostic or therapeutic biopsy of the cervix in a hospital at the presence of histologically confirmed CIN, diagnosed according to the current classification [22,23]. CIN persistence was defined as the histologically confirmed presence of CIN of the same grade upon repeated biopsy or excision within 6–12 months after the initial diagnosis, with no signs of remission during interim clinical or cytological follow-up or cytological verification of low-grade squamous intraepithelial lesion (LSIL)/high-grade squamous intraepithelial lesion (HSIL) (for CIN 1)/HSIL (for CIN 2). The progression of CIN was defined as the increase in the grade of dysplasia (for example, from CIN 1 to CIN 2 or 3) according to repeated histological examination during the follow-up period. None of these patients have been vaccinated.

Follow-up was performed at 6-month intervals using cytology in the first year and annually thereafter. HPV-testing was performed at 6-month intervals in the first year after treatment and annually thereafter.

Clinical data were collected from the patients’ medical records using software in an integrated medical information system. The following clinical data were collected: clinical and pathological data, including patient’s age, body mass index, type of pathology, HPV-strain, subsequent cervical histology, follow-up dates, HPV-strain before treatment, HPV-strain after treatment, and relapse.

The exclusion criteria were as follows: (1) the histological diagnosis other than CIN 1–3 (for example, invasive carcinoma, cervical ectopia); (2) the presence expressed of artifact damage in the area of interest (for example, the absence of the subepithelial stroma < 3 mm); (3) expressed inflammatory or necrotic changes that interfere with morphological assessment; (4) previous destructive cervical therapy (laser vaporization, diathermocoagulation, cryodestruction, or photodynamic treatment) until the time of the test material receiving; (5) lack of subsequent clinical follow-up within ≥6 months after the initial biopsy or excision intervention; (6) unavailability of clinical data on HPV-status or the results of follow-up examinations (cytology, colposcopy, histology); (7) refusal of the patient from further observation or treatment, as well as the severe somatic condition that prevents standard management tactics.

All samples included in this study were independently re-examined by two experienced pathologists (with experience in gynecological pathology). These pathologists acted independently, blindly, and did not know the clinical history. All samples were anonymized prior to the start of the study. The sample size was calculated using standard statistical methods [20] and verified using G*Power software, version 3.1 (Heinrich-Heine-Universität Düsseldorf, Germany) [21].

### 2.2. Histological Examination

All materials were archival cervical specimens (biopsy and surgical) fixed in 10% neutral formalin at 4 °C for 24 h, then washed with tap water and dehydrated in increasing concentrations of alcohol (70%, 90%, 95%, and 100%) and embedded in paraffin blocks. Tissue sections with the thickness of 3 microns were cut using the microtome and placed on the slides. Then, the slides were de-waxed and stained.

*Hematoxylin and eosin staining procedure.* Tissue sections were immersed in Mayer’s hematoxylin for 15 min, and then rinsed with water for 5 min. After that, the sections were stained with eosin for 1 min.

Simple sections stained by hematoxylin and eosin were carefully selected from the tissue block, on which the representative part of the micropreparation was presented.

*The Masson’s trichrome staining procedure.* The commercial kit [Trichrome dye (Masson) Bio-Optica (Milano, Italy)] was used for staining with Masson’s trichrome according to the standard protocol. Collagen fibers were defined as dark blue fibers with black cores. In each series, external control sections were used: dermis (positive control for collagen staining) and internal control—intact cervical stroma.

### 2.3. Histophenotypes of the Extracellular Matrix of the Cervix

The area of interest was a section of subepithelial connective tissue with a depth of at least 3 mm, located directly under the multilayer squamous epithelium in the transformation zone.

Based on morphological assessment and histochemical staining (Masson’s trichrome), three histophenotypes of the subepithelial stroma of the cervix were identified (Figure 1):

The normal histophenotype is characterized by the expressed fibrillar organization: type I collagen fibers form dense parallel-oriented bundles occupying 80–85% of the stroma volume. Stromal fibroblasts are abundantly represented and form the dense net. There is no myxoid component. The cellularity is low and the inflammatory infiltrate is not detected.

The intermediate histophenotype is morphologically characterized by the expressed disorganization of the stromal net, namely, the fibers are thinned, oriented chaotically, with areas of ruptures and focal reduction. There is a focal edema of the fibers; the fibers are fragmented, without clear orientation. A moderate amount of mucoid was detected in the inter-fiber areas. The cellular capacity is significantly increased due to fibroblasts, lymphocytes, and mononuclear elements.

The myxoid histophenotype is characterized by the loss of the organized collagen net and its replacement by the myxoid matrix. Collagen fibers are fragmented. The structure of the stroma is represented by the amorphous, weakly fibrillar myxoid matrix with sections of basophilic mucoid stroma, with the expressed lacunar character, the formation of channels, and the large number of vessels. The inflammatory infiltrate is mainly represented by lymphohistiocytic elements.

Stromal vessels were assessed by hematoxylin and eosin staining in a predefined region of interest located immediately beneath the epithelium. Fields with prominent granulation tissue, coagulation artifacts, or obvious post-manipulation changes were excluded from the assessment. A microvessel was defined as an endothelial-lined lumen distinguishable from the surrounding stroma, with or without red blood cells; vessels with prominent muscular walls and large vessels were ignored. Three random fields of view at 200× magnification were selected by two pathologists (blinded to clinical data) within the region of interest and the average number of vessels was counted. The final result was defined as the average of their morphometric counts.

### 2.4. Ethics Statement

Surplus tissues of the cervix uteri not needed for histopathological diagnosis were selected by experienced pathologists. This study was retrospective, all data were de-identified before analysis, and no intervention in the diagnostic and therapeutic process was performed. The protocol was approved by the Ethics Committee of the Karaganda Medical University (Protocol No. 1 dated 8 January 2025) with the provision of an exemption from individual written consent for a retrospective analysis of de-identified archival materials in accordance with local regulations. This study was performed in agreement with the declaration of Helsinki on the use of human material for research.

### 2.5. Statistical Analysis

Statistical data processing was performed using software Statistica 10.0 (StatSoft Inc., Tulsa, OK, USA) and IBM SPSS Statistics 25.0 (IBM Corp., Armonk, New York, NY, USA). The quantitative variables were initially analyzed using the Shapiro–Wilk test to determine the normality of the distribution and Levene’s test to verify the homogeneity of the variances. For data with normal distribution, the mean, standard deviation (SD), and 95% confidence interval (95% CI) boundaries were calculated. With a normal distribution and homogeneous variances, differences between two independent groups were assessed by Student’s *t*-test; when variances were unequal, Welch’s *t*-test was used. The Mann–Whitney U test was used to compare independent quantitative aggregates in cases where there were no signs of normal data distribution, and the Kruskal–Wallis test was used when comparing more than two groups with post hoc Danna (>2 groups). The comparison of categorical variables between the groups was performed using the χ^2^-test. In the presence of limitations on the sample size, the exact Fisher criterion was applied. The Bonferroni correction was used to control the Type I error rate across multiple comparisons. All tests were two-sided with α = 0.05.

## 3. Results

### 3.1. Comparative Clinical and Morphological Characteristics of the Study Groups

Table 1 shows the clinical and morphological characteristics of the studied groups.

The groups were comparable in age and BMI (mean 33 years and 25 kg/m^2^). Obesity was detected in 18–28% of participants, with no clear trend between the groups. COC use was 18–25% of women, with the proportion of long-term use (>5 years) increasing with more severe epithelial dysplasia: one (14.3%) in the control group, three (33.3%) with CIN 1, three (37.5%) with CIN 2, seven (70%) with CIN 3. The frequency of HPV 16/18 increased with the severity of the lesion: 0% in the control, 17 (42.5%) with CIN 1, 29 (72.5%) with CIN 2, 35 (87.5%) with CIN 3. Inflammation in the stroma was more common in CIN than in the control (67.5–75% versus 32.5%) and was predominantly moderate.

Clinical, demographic, and reproductive characteristics of women with CIN progression/persistence are presented in Table 2.

In a cohort of women with persistent/progressive CIN, the clinical, demographic, and reproductive characteristics were compared between the CIN 1, CIN 2, and CIN 3 groups (age, BMI, age at menarche, number of pregnancies/deliveries, *p* > 0.2). The proportion of patients taking COCs varied from 9.1% in CIN 2 to 50% in CIN 3, with more than 50% of patients taking COCs for more than 5 years. The proportion of patients infected with HPV 16/18 increased with the severity of CIN: one (143%) in CIN 1, three (36.4%) in CIN 2, three (50%) in CIN 3. No histopathological signs of active inflammation were detected; all groups showed no or minimal inflammatory damage with isolated cases of moderate chronic inflammation.

### 3.2. Histopathological Characteristics of Histophenotypes of Cervical Stroma with Cervical Intraepithelial Neoplasia

The distribution of histochemical phenotypes of the cervical stroma was statistically significantly different between groups defined by epithelial dysplasia (CIN 1–3) and by CIN progression/persistence (Figure 2, Table 3).

In the control group, the normal phenotype, characterized by the ordered dense fibrous structure, was determined in 39 (97.5%) cases (95% CI: 87.1–99.6).

There was an intermediate pattern in one case (2.5%). No myxoid changes were detected. In group CIN 1, the normal stroma histophenotype was observed in 11 (27.5%) women (95% CI: 16.1–42.8). The intermediate pattern prevailed in 27 (67.5%) women (95% CI: 52.0–79.9). Myxoid changes were noted in two (5%) cases.

The normal histophenotype was preserved in five (12.5%) patients with CIN 2 (95% CI: 5.5–26.1). The intermediate type was diagnosed in 29 (72.5%) women (95% CI: 57.2–83.9). The myxoid pattern was detected in six (15%) cases.

In patients with CIN 3, the normal histophenotype was not determined in any case (0%). The intermediate type was detected in 18 (45%) women (95% CI: 30.7–60.2), and the myxoid in 22 (55%) (95% CI: 39.8–69.3).

In the subgroup CIN P of repeated biopsy (*n* = 24), which included cases with CIN persistence or progression, the normal histophenotype was preserved in three (12.5%) patients (95% CI: 4.3–31), and the intermediate type in six (25%) (95% CI: 12–44.9). The myxoid phenotype was registered in 15 (62.5%) women (95% CI: 42.7–78.8).

Histopathological signs of inflammation were detected in 17 biopsy results, of which 1 (5.9%) was the normal histophenotype, 10 (58.8%) the intermediate histophenotype, and 6 (35.3%) the myxoid histophenotype. Moderate inflammation was detected in 64 biopsies, of which 4 (6.3%) were the normal histophenotype, 47 (73.4%) the intermediate histophenotype, and 13 (20.3%) the myxoid histophenotype.

The detection rate of HPV 16/18 with the normal histophenotype was 3 (3.7%) cases, with the intermediate histophenotype in 56 (69.1%) cases, and with the myxoid histophenotype in 22 (27.2%) cases.

Quantitative assessment of microvessels demonstrated a directional but statistically insignificant trend towards increasing vessel microdensity from normal to intermediate and then myxoid histophenotype (10.6 ± 4.4, 11.9 ± 5.4, 12.4 ± 5.1, respectively, *p* = 0.545). Age and BMI of women were comparable between groups (Table 3).

## 4. Discussion

The main result of this study was the identification of morphological patterns of the cervical stroma associated with cervical intraepithelial neoplasia of various grades and features of its clinical course. Three reproducible morphological patterns of the extracellular matrix of the cervix have been identified:*The normal histophenotype.* This histophenotype is characterized by the expressed fibrillar organization: type I collagen fibers form ordered parallel-oriented bundles evenly distributed in the stroma. This pattern corresponds to the previously described morphological pattern of the connective tissue of the cervix in the physiological state without inflammation or dysplasia signs [24,25,26].*The intermediate histophenotype.* This histophenotype is represented by areas of disorganization of the collagen fiber net with the change in thickness and local disorientation of the fibers, as well as the appearance of the myxoid component in the inter-fiber space.*The myxoid histophenotype.* This histophenotype is characterized by the violation of the histoarchitectonics of the collagen net with its replacement by the amorphous weakly fibrillar myxoid matrix with sections of basophilic mucoid (myxomatous) stroma.

This study showed that the histochemical picture of the extracellular matrix of the cervix with varying grades of epithelial dysplasia differs at CIN (*p* < 0.05). There was a change in the predominant histophenotype of the cervical stroma as the grade of epithelial dysplasia increased (from CIN 1 to CIN 3) during the transition from normal to intermediate and further to the myxoid type. The control group retained the normal histophenotype with the dense fibrillar structure. The intermediate pattern prevailed at CIN 1 and CIN 2 (67.5% (95% CI: 52.0–79.9) and 72.5% (95% CI: 57.2–83.9%), respectively). The myxoid changes were observed only in 5% of cases at CIN 1 and in six (15%) cases at CIN 2. The most expressed changes in the extracellular matrix of the cervical stroma were registered in group CIN 3 and among patients with confirmed CIN P persistence or progression. In these groups, the myxoid type became dominant (55% (95% CI: 39.8–69.3) and 62.5% (95% CI: 42.7–78.8) respectively). The obtained data indicate a statistically significant association between the myxoid histophenotype of the cervical stroma and more severe grade of epithelial dysplasia, including persistent and progressive forms (*p* < 0.05). Similar morphological patterns have previously been described at neoplastic processes, in which the myxoid transformation of the extracellular matrix correlated with a more aggressive tumor phenotype, decreased immune infiltration, and increased invasion risk [27,28,29,30,31]. We believe that the myxoid histophenotype of the cervical stroma at CIN may reflect pathological precancerous remodeling of the extracellular matrix.

We found further that the HPV 16/18 presence was significantly more often associated with the intermediate (69.1%) and the myxoid (22%) matrix types, whereas HPV 16/18 was detected only in 3.7% of cases at the normal histophenotype. This indicates the possible involvement of oncogenic viruses in the induction of stromal remodeling, regardless of the severity of inflammation. Our data are consistent with current understanding of the role of the microenvironment in CIN persistence [32,33,34,35]. Given the known effect of HPV on the regulation of metalloproteinase expression, as well as its ability to indirectly influence the microenvironment, it can be assumed that viral persistence is one of the key factors in the initiation of morphofunctional destruction of the extracellular matrix at CIN.

The inflammatory process was often associated with the intermediate and myxoid histophenotypes of the extracellular matrix of the cervical stroma. These morphological patterns were accompanied by signs of chronic or subacute inflammation and could reflect reactive remodeling against the background of damage. Interestingly, in a number of cases (up to 23% and 30%, respectively), the intermediate and myxoid phenotypes were registered even in the absence of the express ed inflammatory component or HPV in both the primary biopsy and the CIN progression/persistence. These changes may reflect remodeling of the extracellular matrix of the cervical stroma not only as reactive changes, but also as potentially independent indicators of pathological remodeling; however, given the observational nature of our data and descriptive comparisons, this conclusion is hypothesis-forming and requires confirmation in further multicenter studies with sufficient statistical power.

Additional analysis revealed no significant differences in the age of the patients depending on the histophenotype of the extracellular matrix (*p* > 0.05). These data are consistent with the results of international observational studies, which indicate that the morphological features of dysplasia and stromal remodeling at CIN may be less dependent on the age of patients than on factors of viral persistence, inflammatory background, and molecular disorders of the microenvironment [36].

The key advantage of this study was the comprehensive morphological assessment of the extracellular matrix of the cervical stroma in patients with cervical intraepithelial neoplasia (CIN) of varying severity, including the clinically significant subgroup with persistence and progression of the disease. The stratification of histophenotypes based on Masson’s trichrome staining allowed us to identify reproducible morphological patterns.

The study limitations include fluctuations in data, which may be due to the small number of cases in the subgroups, as well as the lack of long-term follow-up to assess the impact of the identified histophenotypes on the risk of neoplastic progression. In addition, the sample needs to be expanded for stratification by HPV genotypes and types of inflammatory response. Another limitation to be considered is the lack of comparison with immunohistochemical markers and microbiological studies, which would further confirm the origin of the myxoid stroma and significantly expand the understanding of the mechanisms of remodeling. The retrospective study design carries a risk of selection bias, limiting the internal validity of the results. Prospective studies with external validation are needed.

## 5. Conclusions

Histophenotyping of the extracellular matrix of the cervical stroma demonstrates potential as an additional morphological tool for risk stratification in cervical intraepithelial neoplasia. The predominance of the myxoid pattern is associated with CIN 2–3, the persistence of neoplasia, and, probably, with the increased risk of progression. The identified phenotypes can be considered as the tissue equivalent of pathological stromal remodeling in the context of neoplastic transformation at CIN.

Thus, morphological assessment of extracellular matrix patterns has the potential to serve as an additional descriptor to refine risk stratification in primary triage and in postoperative surveillance of CIN persistence/recurrence, helping to clarify the risk of CIN progression, personalize dynamic surveillance, and therapeutic intervention. External prospective multicenter validation, assessment of interobserver reproducibility, and evidence of additional prognostic value relative to current risk factors are required in further studies.

## Figures and Tables

**Figure 1 pathophysiology-32-00055-f001:**
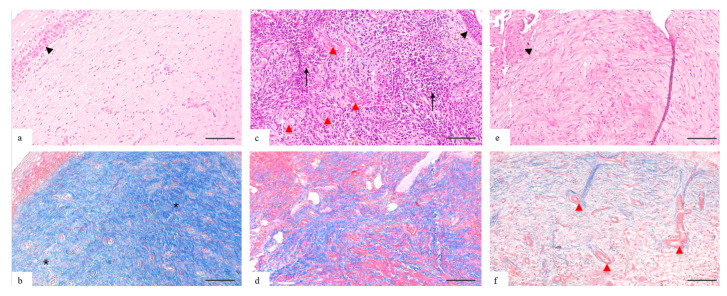
Morphological histophenotypes of the extracellular matrix of the cervical stroma. (**a**,**b**)—Normal histological pattern of the control group. The stroma is represented by compact, parallel-oriented bundles of type I collagen (*) with the uniform distribution of mature fibroblasts. There are no myxoid changes and inflammatory infiltrates. Cellularity is low, inflammatory infiltration is not detected. (**c**,**d**)—Histopathological pattern associated with inflammation and dysplastic lesions of the cervix. Moderate disorganization of the stromal framework: the fibers are thickened, fragmented, oriented chaotically: areas of reduction and rupture are registered. The moderate amount of myxoid tissue is detected between the fibers (*white arrows*). Stroma cellular capacity is significantly increased due to fibroblasts, lymphocytes, and mononuclear cells (*black arrows*). (**e**,**f**)—Histopathological pattern associated with dysplasia: its persistence and progression. The histopathological pattern is represented by the loss of the organized collagen network and its replacement by the myxoid matrix (*white arrows*). Thin-walled vessels are seen in the stroma (*red arrowhead*). (**a**,**c**,**e**)—Hematoxylin and eosin staining, 100× magnification, scale bar 200 μm. (**b**,**d**,**f**)—Histochemical staining of collagen proteins with Masson’s trichrome, 100× magnification, scale bar 200 μm. Symbols: black arrow tip—epithelium; black arrows—cellular infiltration; white arrows—myxoid stroma; *—type I collagen fibers, red arrow tip—microvessels.

**Figure 2 pathophysiology-32-00055-f002:**
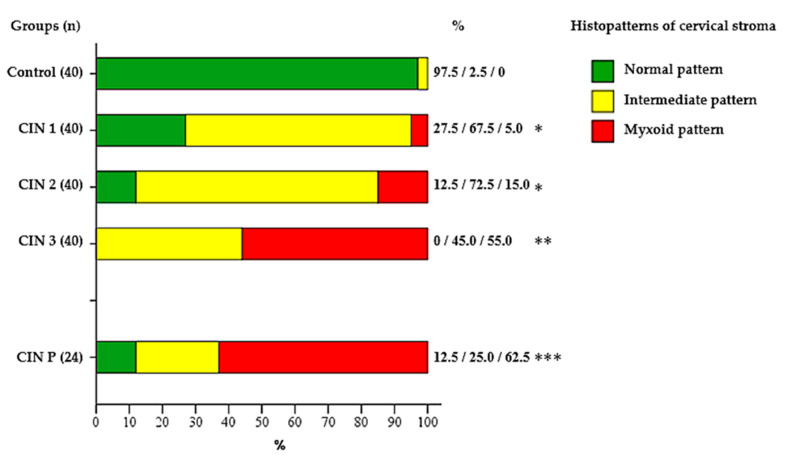
Frequency of cervical stromal extracellular matrix histophenotypes (normal, intermediate, myxoid) in groups defined by epithelial dysplasia (CIN1–3) and among cases with progression/persistence of CIN. Abbreviations: CIN—cervical intraepithelial neoplasia, CIN P—cervical intraepithelial neoplasia with progression/persistence 6 months after treatment or dynamic follow-up *—Statistically significant difference versus of control group (*p* < 0.05); **—Statistically significant difference versus of control group and CIN 1, CIN 2 (*p* < 0.05); ***—Statistically significant difference versus of control group and CIN 1, CIN 2, CIN 3 (*p* < 0.05). Abbreviations: CIN—cervical intraepithelial neoplasia, CIN P—cervical intraepithelial neoplasia with progression/persistence 6 months after treatment or dynamic follow-up.

**Table 1 pathophysiology-32-00055-t001:** Clinical, demographic, and reproductive characteristics of the study group participants at primary biopsy.

Characteristic	Control *n* = 40	With IN 1 *n* = 40	With IN 2 *n* = 40	With IN 3 *n* = 40	*p*-Value
Age, years (average ± SD)	32.68 ± 4.07	33.38 ± 4.67	33.18 ± 4.02	33.88 ± 3.88	0.620
<25, *n* (%)	2 (5)	3 (7.5)	1 (2.5)	-	
26–4 5, *n* (%)	38 (95)	37 (92.5)	39 (97.5)	40 (100)	
>45, *n* (%)	-	-	-	-	
BMI, kg/m^2^ (average ± SD)	25.04 ± 3.53	24.78 ± 2.87	25.16 ± 4.11	24.87 ± 4.24	0.978
Obesity (according to WHO criteria, taking into account race), *n* (%)	11 (27.5)	7 (17.5)	9 (22.5)	10 (25.0)	
Age of menarche onset, years (average ± SD)	12.38 ± 1.17	12.40 ± 1.22	12.48 ± 1.20	12.60 ± 1.26	0.7679
Number of pregnancies (average ± SD)	1.88 ± 1.07	2.13 ± 0.91	2.15 ± 0.83	2.15 ± 0.95	0.0854
Number of births (average ± SD)	1.50 ± 0.88	1.50 ± 0.64	1.95 ± 0.75	1.55 ± 0.71	0.1083
Number of natural births (average ± SD)	1.28 ± 0.88	1.33 ± 0.76	1.78 ± 0.89	1.43 ± 0.81	0.0847
Number of cesarean sections (average ± SD)	0.18 ± 0.38	0.18 ± 0.45	0.25 ± 0.59	0.13 ± 0.46	0.5393
Smoking, *n* (%)					
Yes	-	-	-	-	-
No	40 (100)	40 (100)	40 (100)	40 (100)	
COC reception, *n* (%)					
Yes	7 (17.5)	9 (22.5)	8 (20.0)	10 (25.0)	0.923
No	33 (82.5)	31 (77.5)	32 (80.0)	30 (75.0)	
Duration of COC reception > 5 years, *n* (%)					
Yes	1 (14.3)	3 (33.3)	3 (37.5)	7 (70.0)	0.115
No	6 (85.7)	6 (66.7)	5 (62.5)	3 (30.0)	
Gynecological diseases, *n* (%)					
Adenomatosis	-	-	2 (5.0)	-	0.108
Uterine fibroids	3 (7.5)	-	-	1 (2.5)	0.105
Endometrial polyps	-	2 (5.0)	1 (2.5)	1 (2.5)	0.562
Infections, *n* (%)					
HPV (16/18 strains)	-	17 (42.5)	29 (72.5)	35 (87.5)	**<0.001**
HPV (6/11 and other strains)	6 (15.0)	7 (17.5)	5 (12.5)	4 (10.0)	0.789
HSV	1 (2.5)	3 (7.5)	1 (2.5)	5 (12.5)	0.196
Cytomegalovirus	1 (2.5)	2 (5.0)	1 (2.5)	2 (5.0)	0.875
Other	2 (5.0)	1 (2.5)	2 (5.0)	3 (7.5)	0.789
Chronic diseases, *n* (%)					
Diabetes	-	-	1 (2.5)	-	0.389
Cardiovascular diseases	1 (2.5)	1 (2.5)	1 (2.5)	1 (2.5)	1.00
Arterial hypertension	-	1 (2.5)	2 (5.0)	-	0.292
Histopathological inflammation signs, *n* (%)					
Active acute/subacute/chronic	0/1/2	1/3/2	2/2/4	0/3/0	0.231
Moderate acute/subacute/chronic	1/3/6	4/6/11	1/6/15	2/6/13	
No signs of inflammation/minimal	11/16	9/4	6/4	11/5	

Abbreviations: BMI—body mass index, COC—combined oral contraceptives, WHO—World Health Organization, HPV—human papillomavirus, HSV—herpes simplex virus, SD—standard deviation, *n*—number of cases. The data are presented as Mean ± SD or *n* (%). The tests used were as follows: for normally distributed variables (age, age at menarche)—*t*-test/one-way analysis of variance (Welch’s *t*-test for inequality of variances); for non-normally distributed variables (BMI, pregnancy, childbirth, vaginal delivery, cesarean section)—Mann–Whitney/Krusakal–Wallis test with Dunn–Bonferroni post hoc tests; for categorical—χ^2^ or Fisher’s exact test.

**Table 2 pathophysiology-32-00055-t002:** Clinical, demographic, and reproductive characteristics of women with CIN progression/persistence.

Characteristic	With IN P *n* = 24			*p*-Value
	With IN 1 *n* = 7	With IN 2 *n* = 11	With IN 3 *n* = 6	
Age, years	33.71 ± 5.22	36.36 ± 3.91	37.00 ± 2.19	0.419
<25	-	-	-	
26–4 5	7 (100)	11 (100)	6 (100)	
>45	-	-	-	
BMI, kg/m^2^	24.41 ± 4.99	25.16 ± 4.11	25.75 ± 3.30	0.460
Obesity (according to WHO criteria, taking into account race)	1 (14.3)	2 (18.2)	1 (16.7)	
Age of menarche onset, years	12.43 ± 1.13	11.91 ± 1.14	12.83 ± 1.47	0.391
Number of pregnancies	2.57 ± 1.40	2.64 ± 0.67	2.67 ± 1.21	0.943
Number of births	1.71 ± 0.76	2.36 ± 0.81	2.00 ± 1.10	0.245
Number of natural births	1.43 ± 0.79	2.18 ± 1.17	2.00 ± 1.10	0.220
Number of cesarean sections	0.29 ± 0.49	0.45 ± 0.93	-	0.368
Smoking, *n* (%)				
Yes	-	-	-	-
No	7 (100)	11 (100)	6 (100)	
COC reception, *n* (%)				
Yes	2 (28.6)	1 (9.1)	3 (50.0)	0.171
No	5 (71.4)	10 (90.9)	3 (50.0)	
Duration of COC reception > 5 years, *n* (%)				
Yes	1 (50.0)	1 (100)	2 (100)	-
No	1 (50.0)	-	-	
Gynecological diseases, *n* (%)				
Adenomatosis	-	-	-	-
Uterine fibroids	-	-	-	-
Endometrial polyps	-	-	-	-
Infections, *n* (%)				
HPV (16/18 strains)	1 (14.3)	3 (36.4)	3 (50.0)	-
HPV (6/11 and other strains)	-	1 (9.1)	-	-
HSV	-	-	-	-
Cytomegalovirus	-	-	-	-
Other	-	-	-	-
Chronic diseases, *n* (%)				
Diabetes	-	1 (9.1)	-	-
Cardiovascular diseases	-	1 (9.1)	1 (2.5)	-
Arterial hypertension	-	-	-	-
Histopathological inflammation signs				
Active acute/subacute/chronic	0/0/0	0/0/0	0/0/0	-
Moderate acute/subacute/chronic	0/1/1	0/1/2	0/0/2	
No signs of inflammation/minimal	3/2	3/5	2/2	

Abbreviations: BMI—body mass index, COC—combined oral contraceptives, WHO—World Health Organization, HPV—human papillomavirus, HSV—herpes simplex virus, SD—standard deviation, *n*—number of cases. The data are presented as Mean ± SD or *n* (%). The tests used were as follows: for normally distributed variables (age, age at menarche)—*t*-test/one-way analysis of variance (Welch’s *t*-test for inequality of variances); for non-normally distributed variables (BMI, pregnancy, childbirth, vaginal delivery, cesarean section)—Mann–Whitney/Krusakal–Wallis test with Dunn–Bonferroni post hoc tests; for categorical—χ^2^ or Fisher’s exact test.

**Table 3 pathophysiology-32-00055-t003:** Distribution of clinical and morphological characteristics and inflammatory signs depending on the histophenotype of the cervical stroma.

Characteristic	Normal Pattern *n* = 16	Intermediate Pattern *n* = 74	Myxoid Pattern *n* = 30	*p*-Value
HPV 16/18, *n* (%) *n* = 81	3 (3.7)	56 (69.1)	22 (27.2)	*p* _1_ < **0.001** *p* _2_ < **0.001** *p* _3_ = 0.803
Histopathological signs of inflammation				
Active acute/subacute/chronic	0/1/0	2/3/5	1/4/1	*p* _1_ = **0.002** *p* _2_ = 0.106 *p* _3_ = 0.168
Moderate acute/subacute/chronic	1/2/1	5/11/31	1/5/7	
No signs of inflammation/minimum	8/3	10/7	8/3	
Vessels (average ± SD)	10.56 ± 4.41	11.92 ± 5.35	12.40 ± 5.06	*p* _4_ = 0.545
Age (average ± SD)	33.75 ± 4.42	33.38 ± 4.14	33.57 ± 4.30	*p* _4_ = 0.929
BMI (average ± SD)	23.68 ± 2.10	25.5 ± 4.17	24.20 ± 4.33	*p* _4_ = 0.090

Abbreviations: BMI—body mass index, SD—standard deviation, *n*—number of cases. The data are presented as Mean ± SD or *n* (%). Note: *p* is the significance level; *p* _1_, *p* _2_, *p* _3_—*p*-values of pairwise comparisons between groups; *p*
_1_—normal pattern vs. intermediate pattern, *p*
_2_—normal pattern vs. myxoid pattern, *p*
_3_—intermediate pattern vs. myxoid pattern, *p*
_4_—comparison between three groups: normal pattern, myxoid pattern, and intermediate pattern.

## Data Availability

The data supporting this study’s findings are available on request from the corresponding author.

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
