# Peer review of "Myxoid Stromal Histophenotype Is Associated with High-Grade and Persistent Cervical Intraepithelial Neoplasia"

_pathophysiology, 2025, doi:10.3390/pathophysiology32040055_

Round 1

Reviewer 1 Report

Comments and Suggestions for Authors

The authors of Histophenotypes of the extracellular matrix of the cervix at cervical intraepithelial neoplasia: a retrospective morphological study Diagnostic and prognostic significance of histophenotyping of the extracellular matrix of the cervical stroma at cervical aimed to perform the morphological assessment and comparative analysis of histological patterns of the extracellular matrix of the cervical stroma in normal conditions, at CIN of varying severity, as well as at the persistence or progression of the disease in repeated biopsies.

This single-center retrospective cohort study consistently included biopsy and surgical cervical samples from women of reproductive age with cervical intraepithelial neoplasia (CIN) 1-3.

The samples were examinated for clinical data, histological examination, the Masson's trichrome staining procedure, histophenotypes of the extracellular matrix of the cervix

The main result of the study was the identification of morphological patterns of the cervical stroma associated with cervical intraepithelial neoplasia of various grades and features of its clinical course.

Histophenotyping of the extracellular matrix of the cervical stroma  demonstrates potential as the additional morphological tool for risk stratification in cervical intraepithelial neoplasia. The predominance of the myxoid pattern is associated with CIN 2-3, the persistence of neoplasia and, probably, with the increased risk of progression. The identified phenotypes can be considered as the tissue equivalent of pathological stromal remodeling in the context of neoplastic transformation at CIN.

Kind suggestions:

  1. please mention the quality control of The Masson's trichrome staining procedure.
  2. Try to compare clinical utility of the Masson's trichrome staining procedure with HR HPV genotyping, in stratifying the risk of cervical cancer for women.

Author Response

The authors of Histophenotypes of the extracellular matrix of the cervix at cervical intraepithelial neoplasia: a retrospective morphological study Diagnostic and prognostic significance of histophenotyping of the extracellular matrix of the cervical stroma at cervical aimed to perform the morphological assessment and comparative analysis of histological patterns of the extracellular matrix of the cervical stroma in normal conditions, at CIN of varying severity, as well as at the persistence or progression of the disease in repeated biopsies.

This single-center retrospective cohort study consistently included biopsy and surgical cervical samples from women of reproductive age with cervical intraepithelial neoplasia (CIN) 1-3.

The samples were examined for clinical data, histological examination, the Masson's trichrome staining procedure, histophenotypes of the extracellular matrix of the cervix

The main result of the study was the identification of morphological patterns of the cervical stroma associated with cervical intraepithelial neoplasia of various grades and features of its clinical course.

Histophenotyping of the extracellular matrix of the cervical stroma demonstrates potential as the additional morphological tool for risk stratification in cervical intraepithelial neoplasia. The predominance of the myxoid pattern is associated with CIN 2-3, the persistence of neoplasia and, probably, with the increased risk of progression. The identified phenotypes can be considered as the tissue equivalent of pathological stromal remodeling in the context of neoplastic transformation at CIN.

Kind suggestions:

  1. please mention the quality control of The Masson's trichrome staining procedure.

Answer: Thank you for the comment.

In each series, external control sections were used: dermis (positive control for collagen staining) and internal control – intact cervical stroma. 

  1. Try to compare the clinical utility of the Masson's trichrome staining procedure with HR HPV genotyping, in stratifying the risk of cervical cancer for women.

Answer: Thank you for the comment.

Masson staining is useful as an inexpensive and accessible method to complement routine histology: myxoid stromal pattern is common in CIN3 and in persistence/progression cases, reflecting pathological stromal remodeling. This makes it a promising post-biopsy adjunct to routine histology to clarify prognosis.

However, for population risk stratification and decision making, the “gold standard” is HPV typing, especially 16/18 detection. It is this that provides quantitative risks and is built into protocols for screening and triage of patients.

Prospective multicenter studies are needed to clarify the prognostic value of pathological stromal remodeling in relation to already known risk factors, including HPV 16/18.

Reviewer 2 Report

Comments and Suggestions for Authors

The manuscript by Stabayeva et al. is interesting but does not seem to significantly add to the scientific literature, in my opinion.  That aside, I do feel that the more retrospective data is published, the more information is available within databases to allow researchers to combine data to make predictive models, etc.  Their overall conclusion that histophenotype correlates to the stage of disease (CIN 1-3, P) is just validating the definition as to the categorization of these stages, and not suggesting anything novel.  If this is not the case, then the authors should make clear what their findings mean in the broader context of patient care, especially since the title indicates that these data provide diagnostic and prognostic information.  Finding that HPV16/18 infection significantly more prevalent in patients with later stages of CIN is not novel.  The authors provide no data/evidence to support their statement on lines 441-443:  “Thus, inflammation and viral infection act as potentially synergistic triggers of pathological remodeling of the cervical stroma.”  Additionally, the authors state on lines 443-450, “Interestingly, in the number of cases (up to 23% and 30%, respectively), the intermediate and the myxoid phenotypes were registered even in the absence of the expressed inflammatory component or HPV in both the primary biopsy and the CIN progression/persistence. This allows us to consider these types of extracellular matrix of the cervical stroma not only as reactive changes, but also as potentially independent markers of pathological remodeling of the cervical stroma.” The authors provide no concrete data to support these statements, and they saw varying degrees of inflammation and other changes in each group of patients, and no statistical tests performed on these data in Table 3. Their conclusions (lines 458-470) that first state there is no significant difference in BMI detected among the patients but then state there results suggestively support previous studies as well as stating “…BMI can be considered as the additional modifying factor of the microenvironment at CIN…” are erroneous and should be rephrased.

The manuscript is readable, some minor edits to the English are needed, and the manuscript could be improved in organization and clarity.  I have made some specific suggestions below on these, as well as other edits requested to be made within the manuscript:

  1. The authors have two titles listed on the title page.
  2. Methods section 2.1: many of the paragraphs are single sentences and could be combined into paragraphs. Additionally, the organization could be greatly improved; for example, sentences could be ordered differently so that the groups were outlined and defined before a paragraph on exclusionary criteria, and patients excluded should be in the same paragraph (lines 114-126 and lines 156-158). The two sentences mentioning followup data utilized (lines 146-147 and lines 159-160) should be combined and not separated, and combined with lines 143-145.  Additionally, sample size determination should not be separated from the patients included (lines 143-145) or from how the samples were examined (lines 127-131). 
  3. Methods sections 2.1 and 2.2 could be combined for clarity, but section 2.2 should come before section 2.1 as it defines which patient’s tissues/data were eventually used for analysis.
  4. It wasn’t clear to me whether patient tissue samples were fixed, sectioned, and stained prior to this study, and the authors are re-evaluating these stained slides for this study, or if the authors obtained the tissue samples that were frozen and stored in a tissue bank, upon which the authors sectioned, fixed, and stained each tissue sample. As the authors describe these procedures in detail, I’m assuming the latter, but this should be stated clearly in the methods section. Lines 107-110 states “This single-center retrospective cohort study consistently included biopsy and surgical cervical samples from women of reproductive age with cervical intraepithelial neoplasia (CIN) 1-3. The samples were obtained as the result of diagnostic or therapeutic intervention…” but there is no mention of where samples from normal patients were obtained. Lines 143-145 mention 120 patients with CIN1-3 were used, but what about the other group(s)? They should include that there were 40 control patients in the text as well, since they did so for the other groups.  Easy to state: A total of 160 patient samples were used, 40 control samples and 40 samples each from patients with CIN 1, CIN 2, and CIN 3 (see Table 1).
  5. Line 269: what is meant by “…to reduce the risk of errors of the first kind… “? Are the authors referring to Type I vs Type II error?
  6. Lines 270-272: I’m not sure what is meant by the sentence “When comparing averages, the independent Student's t-test was calculated to compare two independent groups in normally distributed sets of quantitative data.” This seems like a definition of a t-test and is not needed.
  7. Lines 263-272: Along with the previous comment, one potential suggestion would be to state which data are normally distributed vs not, and then state the statistical test used for each data set. This would make this paragraph more readable as well as let readers know what tests were performed on each data set. 
  8. Table 1: The authors could indicate the statistical test(s) used to generate the p-values in the Table legend. As there is only one characteristic that is statistically significant, this can be highlighted, or bolded, for ease of readability.  However, another point that is not addressed, or clear, is how was the correction performed to determine that this one value, out of a list of over 20 characteristics, isn’t just significant due to chance alone?
  9. Just to be more clear, I would suggest referencing Table 2 in line 312, to separate out the data just summarized in Table 1 with the paragraph (lines 312-318) describing Table 2. It would be nice to include in the text in this same paragraph that there was no significant differences in any of the characteristics analyzed among the three groups of patients with CIN P.
  10. I am unclear what data are being analyzed by Figure 2 and subsequently described in the text (lines 348-367). While I understand that the data is graphing the number of patients with either normal, intermediate, or myxoid pattern within the histological section, isn’t this defining control vs CIN patients? I would expect CIN 3 and CIN P to have a higher percentage of myxoid pattern, by definition.  Similarly I would expect control patients to have a normal pattern, by definition, or they wouldn’t be in the control group.  If this isn’t the point of this figure, please clarify.
  11. The first four rows of Table 3 are showing the same data as Figure 2. Are they both needed?
  12. Lines 377-382 are not needed. These data are in Table 3 and appear similar among the groups. The text could state: The age and BMI of the women were similar among groups.
  13. Line 395: histophenotype is incorrectly spelled.
  14. The supplemental figure uploaded is the same as Figure 2. Also, it looks like an original images were uploaded as an additional file but is a duplicate of Figure 1. Are there missing documents? The text doesn’t indicate that there is any supplemental material.
Comments on the Quality of English Language

Minor edits to the English are needed but this can be done fairly easily. 

Author Response

The manuscript by Stabayeva et al. is interesting but does not seem to significantly add to the scientific literature, in my opinion. That aside, I do feel that the more retrospective data is published, the more information is available within databases to allow researchers to combine data to make predictive models, etc. Their overall conclusion that histophenotype correlates to the stage of disease (CIN 1-3, P) is just validating the definition as to the categorization of these stages, and not suggesting anything novel. If this is not the case, then the authors should make clear what their findings mean in the broader context of patient care, especially since the title indicates that these data provide diagnostic and prognostic information.

Answer: Thank you for the comment.

The title of the manuscript has been changed.

The novelty lies in the stromal compartment: we investigate the remodeling of the extracellular matrix of the cervix, characterizing stromal histophenotypes according to Masson's trichrome (including the myxoid pattern) and showing their association not only with the degree but also with the clinical course (persistence/progression) of CIN.

Finding that HPV16/18 infection is significantly more prevalent in patients with later stages of CIN is not novel.

Answer: Thank you for the comment.

We fully agree that the higher frequency of HPV 16/18 in late stages of CIN is a known fact and is not a novelty of our work. In the manuscript, this association is used to test the expected behavior of the cohort relative to published data. The focus of the study is the cervical stromal histophenotype according to Masson as a characteristic of the microenvironment that is not included in the epithelial classification of CIN. We show the association of the stromal phenotype with the severity of the lesion and the persistence/progression of CIN, positioning it as a post-biopsy prognostic method, however, prospective multicenter studies are required to clarify the prognostic value of pathological stromal remodeling relative to already known risk factors, including HPV 16/18 .

We revised the manuscript to emphasize the retrospective and associative nature of the results and the need for external validation.

The authors provide no data/evidence to support their statement on lines 441-443: “Thus, inflammation and viral infection act as potentially synergistic triggers of pathological remodeling of the cervical stroma.”

Additionally, the authors state on lines 443-450, “Interestingly, in the number of cases (up to 23% and 30%, respectively), the intermediate and the myxoid phenotypes were registered even in the absence of the expressed inflammatory component or HPV in both the primary biopsy and the CIN progression/persistence. This allows us to consider these types of extracellular matrix of the cervical stroma not only as reactive changes, but also as potentially independent markers of pathological remodeling of the cervical stroma.” The authors provide no concrete data to support these statements, and they saw varying degrees of inflammation and other changes in each group of patients, and no statistical tests performed on these data in Table 3.

Answer: Thank you for the comment.

We agree that our data do not provide direct evidence of synergy between inflammation and HPV infection; the results obtained are associative. In the revised version, we remove the term “synergistic” and explicitly state that we are talking about observed associations.

To avoid overinterpretation, we have relaxed the wording in the Discussion, adding the notation that the values are descriptive and the conclusions hypothesis-generating. We do not assert the independent marker status of the phenotypes, but rather designate them as potential indicators that require validation in large cohort studies.

Their conclusions (lines 458-470) that first state there is no significant difference in BMI detected among the patients but then state there results suggestively support previous studies as well as stating “...BMI can be considered as the additional modifying factor of the microenvironment at CIN...” are erroneous and should be rephrased.

Answer: Thank you for the comment. The manuscript has been amended.

The manuscript is readable, some minor edits to the English are needed, and the manuscript could be improved in organization and clarity.  I have made some specific suggestions below on these, as well as other edits requested to be made within the manuscript:

  1. The authors have two titles listed on the title page.

Answer: Thank you for the comment. The manuscript has been changed.

In the title of the manuscript : Myxoid Stromal Histophenotype Is Associated with High-Grade and Persistent Cervical Intraepithelial Neoplasia

  1. Methods section 2.1: many of the paragraphs are single sentences and could be combined into paragraphs. Additionally, the organization could be greatly improved; for example, sentences could be ordered differently so that the groups were outlined and defined before a paragraph on exclusionary criteria, and patients excluded should be in the same paragraph (lines 114-126 and lines 156-158). The two sentences mentioning followup data utilized (lines 146-147 and lines 159-160) should be combined and not separated, and combined with lines 143-145. Additionally, sample size determination should not be separated from the patients included (lines 143-145) or from how the samples were examined (lines 127-131).

Answer: Thank you for your detailed comments on the structure of the Materials and Methods section. We have organized and structured the section so that first comes ( 1) study design , location, and observation period; (2) study groups and their definitions; (3) observation protocol; (4) a single paragraph with exclusion criteria.

  1. Methods sections 2.1 and 2.2 could be combined for clarity, but section 2.2 should come before section 2.1 as it defines which patient's tissues/data were eventually used for analysis.

Answer: Thank you for the comment. In the revised version, we have combined 2.1 and 2.2 into a single section.

  1. It wasn't clear to me whether patient tissue samples were fixed, sectioned, and stained prior to this study, and the authors are re-evaluating these stained slides for this study, or if the authors obtained the tissue samples that were frozen and stored in a tissue bank, upon which the authors sectioned, fixed, and stained each tissue sample. As the authors describe these procedures in detail, I'm assuming the latter, but this should be stated clearly in the methods section.

Answer: Thank you for the comment.

All materials were archival cervical specimens (biopsy and surgical) fixed in 10% neutral formalin at 4°C for 24 hours, then washed with tap water and dehydrated in increasing concentrations of alcohols (70%, 90%, 95% and 100%) and embedded in paraffin blocks. Tissue sections of 3 μm thickness were cut with a microtome and placed on glass slides. The slides were then deparaffinized and stained.

Lines 107-110 states “This single-center retrospective cohort study consistently included biopsy and surgical cervical samples from women of reproductive age with cervical intraepithelial neoplasia (CIN) 1-3. The samples were obtained as the result of diagnostic or therapeutic intervention...” but there is no mention of where samples from normal patients were obtained.

Answer: Thank you for the comment.

The control group (defined as the relative physiological reference) in-cluded cervical tissue samples obtained during routine gynecological procedures and autopsies without morphological signs of inflammatory, dysplastic, or neoplastic changes.

Lines 143-145 mention 120 patients with CIN1-3 were used, but what about the other group(s)? They should include that there were 40 control patients in the text as well, since they did so for the other groups. Easy to state: A total of 160 patient samples were used, 40 control samples and 40 samples each from patients with CIN 1, CIN 2, and CIN 3 (see Table 1).

Answer: Thank you for the comment. The manuscript has been amended.

  1. Line 269: what is meant by “...to reduce the risk of errors of the first kind...”? Are the authors referring to Type I vs Type II error?

Answer: Thank you for the comment. The translation in the manuscript has been corrected :

The Bonferroni correction was used to control the Type I error rate across multiple comparisons.

  1. Lines 270-272: I'm not sure what is meant by the sentence “When comparing averages, the independent Student's t-test was calculated to compare two independent groups in normally distributed sets of quantitative data.” This seems like a definition of a t-test and is not needed.

Answer: Thank you for the comment.

We removed the phrase that duplicated the definition of the t - test and rewrote the statistics section more compactly , with clear criteria for choosing parametric/nonparametric methods.

  1. Lines 263-272: Along with the previous comment, one potential suggestion would be to state which data are normally distributed vs not, and then state the statistical test used for each data set. This would make this paragraph more readable as well as let readers know what tests were performed on each data set.

Answer: Thank you for the comment. We have added footnotes to the tables indicating the distribution of variables and the statistical tests used .

  1. Table 1: The authors could indicate the statistical test(s) used to generate the p-values in the Table legend. As there is only one characteristic that is statistically significant, this can be highlighted, or bolded, for ease of readability. However, another point that is not addressed, or clear, is how was the correction performed to determine that this one value, out of a list of over 20 characteristics, isn’t just significant due to chance alone?

Answer: Thank you for the comment. We have highlighted statistically significant differences in bold.

In Table 1, we considered all baseline characteristics for which pooled p-values were available and adjusted for multiple comparisons. We applied the Bonferroni correction at a significance level of α=0.05. The only significant characteristic, HPV 16/18 prevalence, had n < 0.001 and remained significant after adjustment . In addition to p-values, we assessed between-group balance using the effect size (SMD), considering an imbalance of SMD>0.10 to be significant.

  1. Just to be more clear, I would suggest referencing Table 2 in line 312, to separate out the data just summarized in Table 1 with the paragraph (lines 312-318) describing Table 2. It would be nice to include in the text in this same paragraph that there was no significant differences in any of the characteristics analyzed among the three groups of patients with CIN P.

Answer: Thank you for the comment.

We edited the paragraph to add a reference to Table 2 at the beginning of the paragraph to separate it from the description of Table 1. In the same paragraph, we indicated that there were no statistically significant differences between the three groups with IN P (CIN 1, CIN 2, CIN 3) not revealed for any of the analyzed characteristics ( p>0.2 ).

  1. I am unclear what data are being analyzed by Figure 2 and subsequently described in the text (lines 348-367). While I understand that the data is graphing the number of patients with either normal, intermediate, or myxoid pattern within the histological section, isn't this defining control vs CIN patients? I would expect CIN 3 and CIN P to have a higher percentage of myxoid pattern, by definition. Similarly I would expect control patients to have a normal pattern, by definition, or they wouldn't be in the control group. If this isn't the point of this figure, please clarify.

Answer: Thank you for the comment.

Groups C IN 1-3/ Control were formed according to the presence/absence and severity of epithelial pathology ( dysplasia of stratified squamous epithelium ) and clinical course (progression/persistence) ( C IN P ).

Cervical stromal histochemical phenotypes were then assessed independently, blinded, and not used for group allocation. Accordingly, belonging to the control group does not automatically mean a histochemical phenotype of “normal” stroma.

Figure 2 depicts the frequency of stromal histophenotypes within these groups, showing the association between cervical stromal remodeling and epithelial dysplasia/ CIN persistence .

The following changes have been made to the manuscript for clarity:

In methods:

The study groups (CIN 1–3) and CIN P were formed based on epithelial histology (dysplasia of stratified squamous epithelium) and clinical course (progression/persistence of CIN).  Cervical stromal histophenotyping was performed independently, blinded to group status, and was not used for inclusion/exclusion or group assignment.

Caption to Figure 2:

Frequency of cervical stromal extracellular matrix histophenotypes (normal, intermediate, myxoid) in groups defined by epithelial dysplasia ( CIN 1-3) and among cases with progression/persistence of CIN .

The categories on the Y axis represent groups formed by the degree of stratified squamous epithelium dysplasia ( C IN 1-3) and the clinical course (persistence/progression C IN ) (C IN P ); the color indicates the stromal patterns of the extracellular matrix of the cervix.

In results:

The distribution of histochemical phenotypes of the cervical stroma was statistically significantly different between groups defined by epithelial dysplasia (CIN 1-3 ) and by CIN progression/persistence .

  1. The first four rows of Table 3 are showing the same data as Figure 2. Are they both needed?

Answer: Thank you for the comment.

Figure 2 and Table 3 serve complementary purposes: Figure 2 provides a visual comparison of distributions between groups, Table 3 contains the exact values (n, %) for subsequent use of the data.

To eliminate duplication, we removed the first four rows of Table 3 .

  1. Lines 377-382 are not needed. These data are in Table 3 and appear similar among the groups. The text could state: The age and BMI of the women were similar among groups.

Answer: Thank you for the comment. The manuscript has been amended :

Age and BMI of women were comparable between groups .

  1. Line 395: histophenotype is incorrectly spelled.

Answer: Thank you for the comment. The manuscript has been amended .

  1. The supplemental figure uploaded is the same as Figure 2. Also, it looks like an original images were uploaded as an additional file but is a duplicate of Figure 1. Are there missing documents? The text doesn't indicate that there is any additional material.

Answer: Thank you for the comment. We confirm that we do not declare the presence of additional materials in the manuscript and accompanying documents. The uploaded documents are duplicates of the figures and were included in the system by mistake.

Reviewer 3 Report

Comments and Suggestions for Authors

Using current histological staining, the authors categorize three types of cervical stroma in sections from CIN (cervical intraepithelial neoplasia), appropriately illustrated. Lacking is a description of the blood vessels (angiogenesis), clearly visible in the illustrations. The authors find an association between the stromal type and the degree of CIN. The conclusion (L496 to 500) mentions the putative clinical interest of this finding. This statement would  benefit from some further explanation of how the findings might be implemented in the clinic.

A major comment concerns the redaction.  The content of this manuscript is remarkably  redundant. Text and Tables repeat data without new information: See L283 to L311 and Table 1 ; L312 to L 318 and Table 2; L348 to 367 and Figure 2; the first 4 rows of Table 3 and Figure 2; L374 to  L382 and last rows of Table 3. Redundancy is also present in the Discussion.

Three. All abbreviations need explanation when first used in the text, in each figure legend and in footnotes to tables.

Author Response

Using current histological staining, the authors categorize three types of cervical stroma in sections from CIN (cervical intraepithelial neoplasia), appropriately illustrated. Lacking is a description of the blood vessels (angiogenesis), clearly visible in the illustrations.

Answer: Thank you for the comment.

In addition to the added microvessel markers and expanded legend to Figure 1, we have revised the text to describe the method for visually assessing vessels in a standard subepithelial stromal zone and to supplement the results.

The following changes have been made to the manuscript:

In the Materials and Methods section:

Stromal vessels were assessed by hematoxylin and eosin staining in a predefined region of interest located immediately beneath the epithelium. Fields with prominent granulation tissue, coagulation artifacts, or obvious post-manipulation changes were excluded from the assessment. A microvessel was defined as an endothelial-lined lumen distinguishable from the surrounding stroma, with or without red blood cells; vessels with prominent muscular walls and large vessels were ignored. Three random fields of view at ×200 magnification were selected by two pathologists (blinded to clinical data) within the region of interest and the average number of vessels was counted. The final result was defined as the average of their morphometric counts.

In Figure 1, designations have been added to indicate thin-walled vessels, the legend has been expanded, and the description has been supplemented with the following: “Thin-walled vessels are determined in the stroma ( head of the red arrow ).”

In the Results section: Quantitative assessment of microvessels demonstrated a directional but statistically insignificant trend towards increasing vessel microdensity from normal to intermediate and then myxoid histophenotype (10.6 ± 4.4, 11.9 ± 5.4, 12.4 ± 5.1, respectively, p = 0.545).

The authors find an association between the stromal type and the degree of CIN. The conclusion (L496 to 500) mentions the putative clinical interest of this finding. This statement would benefit from some further explanation of how the findings might be implemented in the clinic.

Answer: Thank you for the comment.

Histophenotyping of cervical stromal ECM is a promising, resource-saving additional method for risk stratification in CIN.

In the revised version, we specified that the stromal histophenotype can be considered as an additional descriptor in the pathological report for cervical biopsies both in the initial triage and for the prediction of CIN persistence/recurrence after treatment. In addition, we emphasize that this is a hypothesis-generating area, external prospective validation, assessment of reproducibility and additional predictive value relative to the risk factors already used in prospective multicenter studies are needed.

A major comment concerns the redaction. The content of this manuscript is remarkably redundant. Text and Tables repeat data without new information: See L283 to L311 and Table 1 ; L312 to L 318 and Table 2; L348 to 367 and Figure 2; the first 4 rows of Table 3 and Figure 2; L374 to L382 and last rows of Table 3. Redundancy is also present in the Discussion.

Answer: Thank you for the comment.

We have removed descriptive duplication of data already presented in tables/figures .

Three . All abbreviations need explanation when first used in the text, in each figure legend and in footnotes to tables.

Answer: Thank you for the comment. The text of the manuscript has been amended.

Round 2

Reviewer 2 Report

Comments and Suggestions for Authors

Thank you for addressing my comments. I am satisfied that the manuscript has been revised for clarity and analysis of data to support conclusions.  

Author Response

We sincerely thank you for your positive feedback and for acknowledging the revisions made. We are glad that the changes improved the clarity of the manuscript and strengthened the analysis supporting the conclusions. Your valuable comments greatly contributed to enhancing the quality of our work.

Reviewer 3 Report

Comments and Suggestions for Authors

The authors appropriately answered my comments.

Advice:  accept for publication.

Author Response

We sincerely thank you for your positive evaluation of our manuscript and for recommending it for publication.